# Betel Nut Arecoline Induces Different Phases of Growth Arrest between Normal and Cancerous Prostate Cells through the Reactive Oxygen Species Pathway

**DOI:** 10.3390/ijms21239219

**Published:** 2020-12-03

**Authors:** Li-Jane Shih, Jia-Yu Wang, Jing-Yao Jheng, An-Ci Siao, Yen-Yue Lin, Yi-Wei Tsuei, Yow-Chii Kuo, Chih-Pin Chuu, Yung-Hsi Kao

**Affiliations:** 1Department of Life Sciences, National Central University, Jhongli, Taoyuan 320, Taiwan; shihlijane@gmail.com (L.-J.S.); ericwang2589@yahoo.com.tw (J.-Y.W.); jerry012705@hotmail.com (J.-Y.J.); air200447@gmail.com (A.-C.S.); 2Graduate Institute of Medical Science, National Defense Medical Center, Taipei 114, Taiwan; 3Department of Medical Laboratory, Taoyuan Armed Forces General Hospital, Longtan, Taoyuan 325, Taiwan; 4Department of Emergency, Taoyuan Armed Forces General Hospital, Longtan, Taoyuan 325, Taiwan; er03@aftygh.gov.tw; 5Department of Emergency Medicine, Tri-Service General Hospital, National Defense Medical Center, Taipei 114, Taiwan; 6Department of Gastroenterology, Taiwan Landseed Hospital, Taoyuan 320, Taiwan; YUOYC@landseed.com.tw; 7Institute of Cellular and System Medicine, National Health Research Institutes, Miaoli 350, Taiwan; cpchuu@nhri.org.tw

**Keywords:** areca nut, prostate cancer, cell cycle, cyclin, cyclin-dependent kinase, p21, reactive oxygen species

## Abstract

Prostate cancer (PCa) is a reproductive system cancer in elderly men. We investigated the effects of betel nut arecoline on the growth of normal and cancerous prostate cells. Normal RWPE-1 prostate epithelial cells, androgen-independent PC-3 PCa cells, and androgen-dependent LNCaP PCa cells were used. Arecoline inhibited their growth in dose- and time-dependent manners. Arecoline caused RWPE-1 and PC-3 cell cycle arrest in the G2/M phase and LNCaP cell arrest in the G0/G1 phase. In RWPE-1 cells, arecoline increased the expression of cyclin-dependent kinase (CDK)-1, p21, and cyclins B1 and D3, decreased the expression of CDK2, and had no effects on CDK4 and cyclin D1 expression. In PC-3 cells, arecoline decreased CDK1, CDK2, CDK4, p21, p27, and cyclin D1 and D3 protein expression and increased cyclin B1 protein expression. In LNCaP cells, arecoline decreased CDK2, CDK4, and cyclin D1 expression; increased p21, p27, and cyclin D3 expression; had no effects on CDK1 and cyclin B1 expression. The antioxidant N-acetylcysteine blocked the arecoline-induced increase in reactive oxygen species production, decreased cell viability, altered the cell cycle, and changed the cell cycle regulatory protein levels. Thus, arecoline oxidant exerts differential effects on the cell cycle through modulations of regulatory proteins.

## 1. Introduction

Prostate cancer (PCa) is the most common reproductive system cancer in elderly men. The American Cancer Society estimated that 180,890 new cases of PCa occurred in the US in 2016 and that approximately 26,120 deaths occurred that same year from PCa-specific disease. Generally, early PCa has no clear symptoms, while advanced PCa commonly develops and spreads from the prostate to the bones and causes bone pain and urinary dysfunction [1]. Although androgen deprivation, which was discovered by Charles Huggins in 1941, can cause the regression of androgen-responsive metastatic PCa, hormone therapy eventually induces the development of castration-resistant PCa within 3 years [2]. As the growth and development of PCa are regulated by food compounds [2,3], it is worthwhile to carefully examine the signaling pathways through which such compounds differentially affect the growth of normal prostate cells, androgen-independent PCa cells, and androgen-dependent PCa cells.

Betel nut alkaloids (BNAs), especially arecoline, have been reported to be regulatory agents of the prostate gland in male rats [4]. Specifically, in one study, arecoline treatment increased the weight of the prostate gland and serum levels of gonadotropin and testosterone hormones. Elevated androgen levels induced by arecoline may lead to increased cell proliferation in the prostate gland, as indicated by increased levels of Ki-67 protein and regulatory proteins of the G1-to-S cell cycle, such as cyclin D1 and cyclin-dependent kinase (CDK) 4. Various in vivo studies in rats have shown that arecoline may also stimulate Leydig cell activity and the release of testosterone possibly via the inhibition of pineal activity [5]. Despite this evidence that arecoline has an indirect impact on the rat prostate gland through the modulation of testosterone levels, it is still unknown whether arecoline exerts direct, differential effects on normal and cancerous human prostate cells. Unfortunately, the results of these studies [4,5] did not demonstrate the effect of either guvacine or arecaidine on the growth of normal and cancerous prostate cells. Notably, arecoline differed from guvacine and arecaidine in inhibiting preadipocyte growth with reactive oxygen species (ROS) dependency [6]. Therefore, it is worthwhile to explore whether arecoline or other BNAs differentially affect the growth of normal prostate cells, androgen-independent PCa cells, and androgen-dependent PCa cells.

Arecoline regulates the growth of non-prostate cells in part through cell cycle regulatory proteins and reactive oxygen species (ROS) production [6,7,8,9,10]. Specifically, arecoline was found to induce dysregulation of the G1 or G2 phase of the cell cycle in non-PCa cells through the modulation of cell cycle-related proteins (e.g., p21 and CDKs) and ROS. Despite these findings, no studies have demonstrated whether any cell cycle regulatory proteins, such as CDK1, CDK2, CDK inhibitors (CKIs), and cyclins, can transduce anti-growth signals induced by arecoline to regulate the cell cycle in human PCa cells. Further studies are therefore necessary to determine whether the oxidant activity of arecoline is required for its differential effects on the growth of normal prostate cells, androgen-independent PCa cells, and androgen-dependent PCa cells. These studies may help elucidate the mechanism that underlies the actions of arecoline on the prostate.

This study was designed to increase our understanding of the mechanism by which betel nut arecoline differentially regulates the growth of normal RWPE-1 prostate epithelial cells, androgen-independent human PC-3 PCa cells, and androgen-dependent human LNCaP PCa cells. We confirmed that arecoline, but not arecaidine or guvacine, significantly reduced the viability of all three cell lines. We showed that high doses of arecoline differentially induced cell cycle dysregulation among RWPE-1, PC-3, and LNCaP cells, and selectively affected specific proteins in the CDK, CKI, and cyclin families in a cell type-dependent manner. We also found that the effects of arecoline are dependent on pathways that require intracellular ROS production.

## 2. Results and Discussion

### 2.1. Effects of BNAs on Cell Viability

Significant variations were observed in the measured cell viability (Figure 1). In RWPE-1, PC-3, and LNCaP cells, arecoline was generally more effective in reducing cell viability than arecaidine or guvacine (Figure 1A–C), and the effects depended on the dosage and duration of treatment. For example, the IC_50_ values of arecaidine and guvacine for all three cell lines were all greater than 1 mM during 48 h of treatment, except for the approximate value of 0.45 mM of arecoline in the treatment of RWPE-1 and PC-3 cells at 48 h. The IC_50_ values of arecoline in the treatment of LNCaP cells were 0.8–1 mM at 48 h. These observations suggest the alkaloid-specific effect of betel nut on the growth of normal and cancerous prostate cells.

### 2.2. Arecoline-Induced Differential Growth Arrest in Normal and Cancerous Prostate Cells

Alterations in the cell cycle can modulate cell growth [11]. We found that arecoline changed the percentages of normal RWPE-1 cells in the G0/G1, S, and G2/M phases of the cell cycle (Figure 2 and Appendix A). Generally, treatment with arecoline at a concentration of 0.4 mM for 48 h tended to decrease the percentage of cells in the G1 and S phases and tended to increase the percentage of cells in the G2/M phase (Figure 2A). Similar effects were observed when PC-3 cells were treated with arecoline, in that the percentage of cells in the G1 phase tended to decrease, whereas the percentage of cells in the G2/M phase tended to increase (Figure 2B). However, in LNCaP cells, arecoline increased the percentage of cells in the G1 phase and decreased the percentage of cells in the S phase, but had no significant effect on the percentage of cells in the G2/M phase (Figure 2C). These observations suggest that the effects of arecoline on the cell cycle of normal and cancerous prostate cells are cell type dependent. This induced dysregulation of the cell cycle by arecoline, which is supported by altered percentages of cells in the G1, S, and G2/M growth phases, may explain the observed decreases in the viability of RWPE-1, PC-3, and LNCaP cells treated with arecoline. The results obtained in RWPE-1 and PC-3 cells are similar to those reported for the G2/M-dependent effect of arecoline on mucosal fibroblasts, vascular endothelial cells, adipose cells, and basal cell carcinoma cells [6,7,9,12], but the results obtained in LNCaP cells are consistent with those reported for the G0/G1-dependent effects of arecoline on normal hepatocytes and keratinocytes [8,10].

### 2.3. Arecoline Selectively Affected Specific Types of Cell Cycle-Regulating Proteins

The cell cycle can be regulated by proteins in the CDK, CKI, and cyclin families [11]. For example, CDK2 and p21 were found to regulate PCa cell growth in the absence of arecoline [13]. In other experiments found in non-prostate cells, p21 mediated areoline-induced growth arrest of hepatocytes [8] and keratinocyte [10], CDK1 mediated arecoline-induced G2/M arrest of basal cell carcinoma cells [12], and CDK1 and cyclin B2 acted as signaling components in pathways through which arecoline mediates the cell cycle of 3T3-L1 preadipocytes [6]. In our experiment, RWPE-1 cells treated with arecoline at a concentration of 0.4 mM, but not at concentrations of 0.05 and 0.2 mM, exhibited a significant increase in the CDK1 protein expression levels and exhibited a tendency for decreased CDK2 protein expression, whereas arecoline had no effect on CDK4 protein expression after 24 h of treatment (Figure 3A). In PC-3 cells, arecoline tended to decrease the expression of CDK1, CDK2, and CDK4 proteins in a dose-dependent manner (Figure 3B). In LNCaP cells, arecoline did not alter the CDK1 protein level, but it reduced the CDK2 and CDK4 protein levels in a dose-dependent manner (Figure 3C). Since RWPE-1, PC-3, and LNCaP cells represent normal prostate epithelial cells, androgen-independent PCa cells, and androgen-dependent PCa cells, respectively, these observations suggest that the selective effects of arecoline on specific proteins of the CDK family are differentially induced among these cell types.

When the CKI family was examined, we found that arecoline at concentrations of 0.05 and 0.2 mM had no effect on p21 protein expression in RWPE-1 cells. However, when 0.4 mM arecoline was used, a significant increase in the p21 expression level was observed (Figure 4A), whereas treatment with 0.4 mM arecoline did not alter the p27 protein expression level. In PC-3 cells, 0.2 and 0.4 mM, but not 0.05 mM, arecoline tended to decrease the p21 and p27 protein expression levels (Figure 4B). In LNCaP cells, treatment with 0.4 mM arecoline induced increases in the expression of both p21 and p27 proteins (Figure 4C).

When the cyclin family was examined, we observed that arecoline at concentrations of 0.05 and 0.2 mM had no effect on the levels of cyclins B1, D1, and D3, but that 0.4 mM arecoline significantly increased the levels of cyclins B1 and D3 (Figure 5A). In PC-3 cells, treatment with 0.4 mM arecoline increased the levels of cyclin B1 protein and decreased the levels of cyclin D1 and D3 proteins (Figure 5B). In LNCaP cells, treatment with 0.4 mM arecoline did not induce significant changes in the levels of cyclin B1 protein, but the levels of cyclin D1 were significantly decreased, while the levels of cyclin D3 were increased (Figure 5C).

### 2.4. The Effects of Arecoline on Normal and Cancerous Prostate Cell Growth Was Dependent on the ROS Pathway

Although the anti-growth effect of arecoline on human cells has been reported to be ROS dependent [7,8,9,10], no published studies were found on normal human prostate and PCa cells. Thus, we examined whether the arecoline-induced changes in cell viability, growth arrest phase, and cell cycle regulatory molecules in RWPE-1, PC-3, and LNCaP cells were dependent on the ROS pathway (Figure 6, Figure 7 and Figure 8). Indeed, we first found that arecoline induced a significant increase in ROS production in all three cell types after 24 and 48 h of treatment (Figure 6). Pretreatment with N-acetylcysteine (NAC) blocked the arecoline-induced increases in the levels of ROS production in all cells (Figure 6). In addition, NAC antagonized the arecoline-induced decreases in the viability of RWPE-1, PC-3, and LNCaP cells (Figure 7). In the presence of arecoline, NAC antagonized the arecoline-altered percentages of RWPE-1, PC-3, and LNCaP cells in the G1, S, and G2/M phases (Figure 2). Interestingly, NAC alone induced increases in the percentage of RWPE-1 cells in the G1 phase, decreases in the percentage of those cells in the S phase, and induced no changes in the percentage of those cells in the G2/M phase (Figure 2A). In PC-3 and LNCaP cells, NAC alone did not affect the percentages of cells in the G1, S, and G2/M phases.

Next, we examined whether the arecoline-induced alterations in the levels of cell cycle regulatory proteins were affected by NAC (Figure 8, Figure 9 and Figure 10). In RWPE-1 cells, pretreatment with NAC prevented arecoline-induced alterations in the levels of CDK1, CDK2, cyclin B1, and cyclin D3 proteins (Figure 8). Although NAC did not significantly block arecoline-induced increases in p21 protein levels, it slightly antagonized the effect of arecoline. Interestingly, in RWPE-1 cells, NAC alone tended to decrease the levels of CDK1, cyclin B1, and cyclin D3 proteins and increased the levels of p21, while it had no effect on the levels of CDK2 (Figure 8). In PC-3 cells, NAC alone did not have any effects on the levels of CDK1, CDK2, p21, and cyclin B1 proteins (Figure 9). However, arecoline treatment prevented arecoline-induced alterations in the levels of CDK2, p21, and cyclin B1 proteins and had no effect on the arecoline-increased levels of CDK1 protein. In LNCaP cells, NAC alone did not induce significant changes in the levels of CDK2, CDK4, p21, p27, cyclin D1, or cyclin D3 proteins (Figure 10). In the presence of arecoline, NAC prevented the arecoline-induced decrease in the levels of CDK2, CDK4, and cyclin D1 proteins and the arecoline-induced increase in the levels of p21, p27, and cyclin D3 proteins. Taken together, these observations suggest that the cell type-dependent effect of arecoline on prostate cell cycle regulatory proteins is mediated through a pathway that requires the induction of ROS production. It was evident that arecoline could participate in the generation of different metabolites and free radical groups, such as arecoline N-oxide, arecoline N-oxide mercapturic acid, nitrosamines, 3-methylnitrosaminopropionate, and N-nitrosoguvacoline [14,15]. Whether this mechanism explains the ROS-dependent effect of arecoline on normal and cancerous prostate cell cycle regulatory proteins was not demonstrated in this study.

Generally, CDK1 and CDK2/CDK4/CDK6 control the checkpoints of the G2/M phase and the G0/G1 phase, respectively, p21 and p27 act as endogenous inhibitors of CDKs, and cyclin B1 and cyclin D function as the endogenous activators of CDK1 and CDK4/6, respectively [16,17]. Increased p21 protein expression in RWPE-1 cells may inhibit CDK2 activity and lead to significant growth arrest in the G2/M phase. Cyclin B1 is a G2/M cyclin that is associated with the CDK1 protein and, when it accumulates, favors M-phase arrest at the G2/M checkpoint [11,18]. Therefore, the observed increase in cyclin B1 protein levels after treatment with 0.4 mM arecoline for 24 h strengthens the possibility of CDK1-related effects of arecoline on RWPE-1 growth arrest at the G2/M phase. The blocking effect of NAC on arecoline-induced increases in both CDK1 and cyclin B1 proteins helps explain the progression of cells from the G2/M phase to the G1 phase. Concurrently, the high p21 protein levels observed in the combined NAC and arecoline treatment group favor RWPE-1 cell arrest in the G1 phase, which was observed at 24 and 48 h. As p21 and p27 are endogenous proteins that favor cell cycle arrest of human PCa cells at the G1 checkpoint [13], decreased expression levels of p21 and p27 proteins in PC-3 cells by arecoline treatment may lead to significant decreases in the percentage of cells in the G0/G1 phase. Compared with arecoline alone, the blocking effect of NAC on the arecoline-induced decreases in CDK2 and p21 proteins may help rescue the percentage of PC-3 cells in the G1 phase after 24 and 48 h of treatment. Since CDK1 becomes the predominant CDK in the early G2/M transition of the cell cycle, and since cyclin B1 is the M-phase arrest protein [11,16,17,18], decreased CDK1 protein levels and increased cyclin B1 protein levels may result in the G2/M growth arrest in PC-3 cells, which was observed 24 and 48 h after arecoline treatment. The blocking effect of NAC on the arecoline-altered levels of cyclin B1, but not CDK1, may explain the lower percentage of cells in the G2/M phase in the combined NAC and arecoline group relative to the arecoline-only group. This phenomenon may also indicate the involvement of high cyclin B1 protein levels as arecoline exerts its effects. These findings are consistent with those reported for the arecoline-induced G2/M-phase arrest in KB human epidermoid carcinoma cells [18,19]. However, in LNCaP cells, decreased CDK2, CDK4, and cyclin D1 protein levels and increased p21, p27, and cyclin D3 protein levels induced by arecoline appear to favor cell arrest in the G1 phase. The finding that arecoline treatment did not lead to alterations in the CDK1 and cyclin B1 protein levels helps explain the lack of significant changes in the percentage of cells in the G2/M phase. The blocking effect of NAC on the arecoline-altered levels of CDK2, CDK4, p21, p27, cyclin D1, and cyclin D3 helps explain the lower percentage of cells in the G1 phase in the combined NAC and arecoline treatment group relative to the arecoline-only group; this also indicates the involvement of these proteins in the effects of arecoline on the G1 growth arrest of androgen-dependent PCa cells.

Arecoline was found to possess multiple functions, including effects on endocrine, nervous, gastrointestinal, and cardiovascular systems [20]. This could be explained by various molecule targets of arecoline, including muscarinic receptors, anti-atherogenic factors (e.g., nitric oxide), inflammatory cytokines, calcium channels, steroid dehydrogenase, pancreas-duodenum homeobox-1, heme oxygenase-1, cell cycle-related proteins (e.g., CDK, CKI, and cyclins), the phosphatidylinositol-3-kinase (PI3K)–mammalian target of rapamycin (mTOR)–p53 pathway, catenin, and so forth. Although the exact target of arecoline in normal and cancerous prostate cells was not demonstrated in this study, we provided a certain in-depth understanding of its effects on the protein expression of particular types of CDK, CKI, and cyclin family members with ROS dependency. It is worthwhile to explore further whether any of those reported arecoline target molecules are necessary for its stimulating ROS production and altering levels of CDK, CKI, and cyclin proteins in normal and cancerous prostate cells.

Differences of androgen sensitivity and tumorigenic activity occur among RWPE-1, PC-3, and LNCaP cell lines [21,22]. Although RWPE-1 cells were immortalized with human papilloma virus, they are non-tumorigenic and androgen sensitive with positive androgen receptor (AR) and prostate-specific antigen (PSA) expression. The cell line is positive for phosphatase and tensin homolog (PTEN), retinoblastoma (Rb), and p53 proteins, as well as response to epidermal growth factor (EGF) and transforming growth factor. PC-3 cells isolated from a vertebral metastatic prostate tumor are androgen insensitive without AR or PSA expression, and they are PTEN deficient and express high levels of EGF receptor (EGFR), Rb protein, and aberrant p53. LNCaP cells isolated from a human lymph node metastatic lesion of prostate adenocarcinoma are androgen sensitive with AR and PSA expression, and they are PTEN deficient and express EGFR protein and a low level of Rb protein. In our study, arecoline induced different phases of growth arrest between normal and cancerous prostate cells through the ROS pathway, as well as causing different alteration patterns of CDK, CKI, and cyclin protein levels. As AR, PTEN, p53, and Rb proteins can function as regulators of cell cycle progression and proliferation of prostate cells and they are differentially expressed among RWPE-1, PC-3, and LNCaP cells [21,22], we could not exclude the possibility that any of them or other protein signaling cascades (e.g., EGFR and AKT) might be involved in the distinct effects of arecoline on the growth and cell cycle proteins of RWPE-1, PC-3, and LNCaP cells. Notably, androgen-dependent LNCaP cells exhibited the different protein expression profiles and chemotherapy drug responses from androgen-independent PC-3 cells [22]. 

In vivo, arecoline was reported to increase the weight of the prostate gland and serum levels of gonadotropin and testosterone hormones and thereby leading to increased cell proliferation in the prostate gland [4]. It was evident by increased levels of Ki-67 protein, cyclin D1, and CDK4 and by increased activity of Leydig cells [4,5]. Interestingly, our study indicated that arecoline inhibited the growth of normal prostate cells, androgen-independent PCa cells, and androgen-dependent PCa cells. These observations were supported by an increased percentage of the G2/M phase in the RWPE-1 and PC-3 cell cycle and by an increased percentage of the G0/G1 phase arrest in the LNCaP cell cycle. In addition, in RWPE-1 cells, arecoline increased levels of CDK-1, p21, and cyclins B1 and D3, decreased CDK2 protein levels, and had no effects on CDK4 and cyclin D1 protein levels. In PC-3 cells, arecoline decreased CDK1, CDK2, CDK4, p21, p27, and cyclin D1 and D3 protein expression and increased cyclin B1 protein expression. In LNCaP cells, arecoline decreased CDK2, CDK4, and cyclin D1 expression, increased p21, p27, and cyclin D3 expression, and had no effects on CDK1 and cyclin B1 expression. A possible explanation for the in vivo and in vitro discrepancy is that the action of arecoline on prostate cell growth varies with the presence of testosterone and other serum factors, the varying levels and types of downstream signaling proteins (e.g., CDKs, CKI, and cyclins), and prostate cell types. Thus, arecoline may not only have an indirect impact on the growth of prostate cells through the modulation of serum factor levels, but also exert a direct effect on the proliferation of normal and cancerous human prostate cells.

BNAs have numerous biological properties that can potentially result in various biological effects [4,5,6,7,8,9,10,18,19,23,24,25,26,27,28,29,30,31,32,33,34]. In most cases, but not all, arecoline is more active than other alkaloids. This contention is supported by our findings in RWPE-1, PC-3, and LNCaP cells because at the same dose and duration of treatment, arecoline was generally more effective than arecaidine and guvacine in altering the viability of these cells. The observed alkaloid-specific effects of the betel nut suggest that arecoline may act differently from arecaidine and guvacine in regulating the growth of normal and cancerous prostate cells. According to the nature of the unique structures of these three alkaloids tested [29], arecoline, but not arecaidine or guvacine, contains a methyl ester group on its N-containing aromatic ring. This methyl ester group of arecoline may be important for some conformational flexibility and in its interactions with other molecules.

## 3. Materials and Methods

### 3.1. Chemical Reagents

All active reagents (e.g., arecoline hydrobromide, arecaidine hydrochloride, guvacine hydrochloride, and others) were purchased from Sigma Chemical (St. Louis, MO, USA), unless otherwise stated. BNAs were dissolved in sterile medium for cell treatment. RPMI-1640, Keratinocyte-SFM, fetal bovine serum (FBS), trypsin, and protein markers were purchased from Gibco-Invitrogen (Grand Island, NY, USA). The CDK4, CDK6, cyclin B1, cyclin D1, cyclin D3, p21, p27, and actin antibodies were obtained from Cell Signaling Technology (Billerica, MA, USA), while other antibodies (i.e., CDK1, CDK2, and all others) were purchased from Santa Cruz Biotechnology (Santa Cruz, CA, USA) (Table 1).

### 3.2. Cell Culture

According to a previously published method [13], RWPE-1 cells (American Type Culture Collection, Manassas, VA, USA; CRL-11609), PC-3 (ATCC, CRL-1435), and LNCaP-FGC (ATCC, CRL-1740) were seeded at a density of approximately 10,000–20,000 cells/cm^2^. RWPE-1 cells were grown in Keratinocyte-SFM (pH 7.4) containing 0.05 mg/mL bovine pituitary extract (Gibco), 5 ng/mL human epidermal growth factor (EGF), 100 μg/mL penicillin, and 100 μg/mL streptomycin in a humidified atmosphere of 95% air and 5% CO_2_ at 37 °C. PC-3 and LNCaP-FGC PCa cells were grown in RPMI-1640 supplemented with 10% FBS, 100 μg/mL penicillin, and 100 µg/mL streptomycin (GibcoBRL) in a humidified atmosphere of 95% air and 5% CO_2_ at 37 °C.

### 3.3. Cell Viability

To determine whether BNAs exert their effects on RWPE-1, PC-3, and LNCaP cell viability in a dose- or time-dependent manner, cells (5000 cells/well in a 96-well plate) were treated with various concentrations (0–1 mM) of arecoline, arecaidine, or guvacine in the presence of 10% FBS-supplemented medium for the indicated time periods [6,35]. After a particular duration in culture, 0.5 μg/mL of the tetrazolium dye (3-(4,5-dimethylthiazol-2-yl)-2,5-diphenyltetrazlium bromide [MTT]) were added to the cells, which were incubated in the dark at 37 °C for 3 h. Then, an aliquot of 100 µL of 100% dimethyl sulfoxide (DMSO) was added to stop the reaction; this step allowed the insoluble formazan to dissolve in DMSO. The absorbance was then read at 570 nm.

### 3.4. Experimental Treatment

We followed the methods reported by Tien et al. [6] and Wang et al. [36] to investigate the ROS-dependent effect of arecoline. RWPE-1, PC-3, and LNCaP cells were pretreated with the antioxidant N-acetylcysteine (NAC; 1.5 mM) for 1 h and then exposed to 0.4 mM arecoline. After 24 or 48 h of treatment, we measured cell viability, the percentage of cells in different phases of the cell cycle, the expression of cell cycle-related proteins, and ROS production.

### 3.5. Flow Cytometric Analysis

Changes in cell cycle kinetics were analyzed by flow cytometry, as described by Hung et al. [37]. RWPE-1, PC-3, and LNCaP cells were seeded at a density of 6 × 10^5^ cells/plate in a 10 cm dish, were treated with or without NAC for 1 h, and then exposed to 0.4 mM arecoline. After 24 and 48 h, cell pellets were harvested, fixed in 70% ethanol, and stored at −20 °C until analysis. For analysis, the cell pellets were washed with 10 mM cold phosphate-buffered saline (PBS) (pH 7.4), incubated at 37 °C for 30 min with 100 μg/mL RNase A, and then stained with 200 μg/mL propidium iodide in PBS containing 1% Triton X-100. The cell cycle profiles and distributions were determined by flow cytometric analysis of 10^4^ cells using the CELLQuest program on a FACSCalibur^™^ flow cytometer (Becton Dickinson, San Jose, CA, USA). Gating was used to exclude clumped cells from the cell cycle distribution analysis.

### 3.6. Western Blot Analysis

Western blot analysis was performed on supernatant fractions of RWPE-1, PC-3, and LNCaP cells, as described by Shih et al. [38]. An aliquot of approximately 50–75 μg of supernatant protein was separated by 12% sodium dodecyl sulfate (SDS)-polyacrylamide gel electrophoresis with 2× gel-loading buffer (100 mM Tris-HCl, pH 6.8, 4% SDS, 20% glycerol, 0.2% bromophenol blue, and 10% β-mercaptoethanol) and then blotted onto Immobilon-NC transfer membranes (Millipore, Bedford, MA, USA). The immunoblots were blocked for 1 h at room temperature with 10 mM PBS containing 0.1% Tween 20 and 5% skim milk. The primary and secondary antibodies were used at dilutions of 1:1000 (~0.2 μg/mL) and 1:2000 (~0.2 μg/mL), respectively. The immunoblots were visualized using the Western Lightning^™^ Chemiluminescence Reagent Plus kit (PerkinElmer Life Science, Boston, MA, USA) for 3 min, followed by exposure to Fuji film for 2–3 min. Blots were quantified using the FX Pro Plus Molecular Imager^®^ (Bio-Rad Laboratories, Hercules, CA, USA). After normalization to β-actin protein expression, the levels of these intracellular proteins were expressed as a percent of the control, unless noted otherwise.

### 3.7. ROS

For the ROS assay, we followed the methods reported by Tien et al. [6] and Wang et al. [36]. RWPE-1, PC-3, and LNCaP cells (1.2 × 10^5^ cells/well in a six-well plate) were pretreated with NAC for 1 h and then incubated with 0.4 mM arecoline. After 24 or 48 h of incubation, 30 μM of 2′,7″-dichlorofluorescein diacetate (DCFDA) was added. After a 30 min incubation in the dark at 37 °C, dichlorofluorescein generated from the reaction of DCFDA with ROS was detected by fluorescence emission using a fluorescence spectrophotometer (F-4500, HITACHI) under an excitation wavelength of 504 nm and emission wavelength of 529 nm. Based on the absorbance value, the ROS levels were normalized to the number of cells and then expressed as multiples relative to the control. Cells were trypsinized and counted using the 0.4% trypan blue exclusion method. Only live cells were represented in this study.

### 3.8. Statistical Analysis

We presented the data as the mean ± SEM. A statistical analysis was performed, as described by Shih et al. [39].

## 4. Conclusions

We concluded that the effects of arecoline on the growth of normal RWPE-1 and cancerous PC-3 and LNCaP prostate cells are likely mediated through cell cycle dysregulation (Figure 11). The effects are related to the induction of different phases of growth arrest as well as alterations in the levels of specific members of the CDK, CKI, and cyclin protein families and are induced in a ROS-dependent manner. Arecoline is more effective than arecaidine and guvacine in the induction of prostate cell growth changes. Since the acetylcholine receptor (AchR) can function as an arecoline receptor [24,29], further studies are needed to determine whether AchR is responsible for the observed effects of arecoline on normal and cancerous prostate cells.

## Figures and Tables

**Figure 1 ijms-21-09219-f001:**
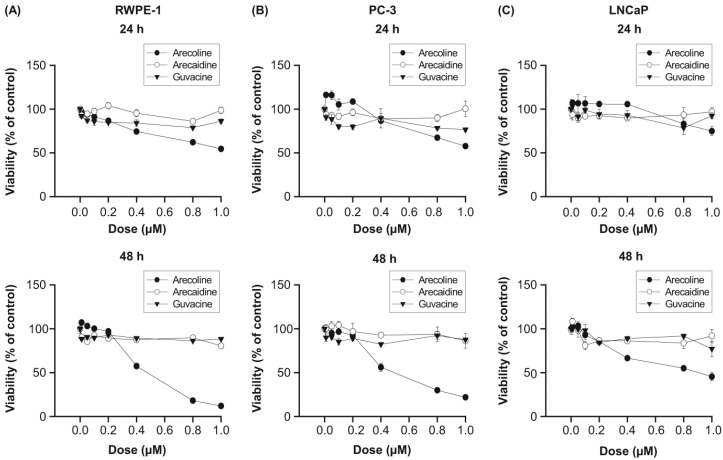
Reducing effect of betel nut alkaloids on the viability of normal RWPE-1 (**A**), cancerous PC-3 (**B**), and LNCaP (**C**) prostate cells with various concentrations (0–1 mM), duration (24–48 h) of treatment, and with different types of alkaloids. Data are expressed as the mean ± SEM from triplicate experiments. For clarity, statistical significance is not shown.

**Figure 2 ijms-21-09219-f002:**
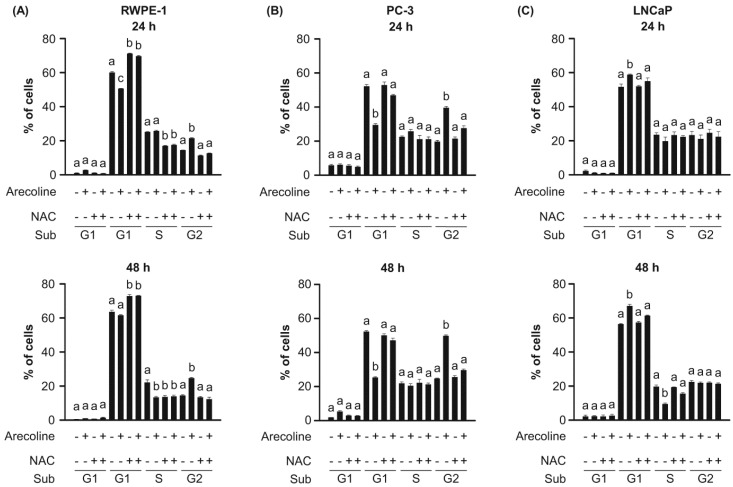
Arecoline-induced growth arrest at different phases of the cell cycle in normal RWPE-1 (**A**), cancerous PC-3 (**B**), and LNCaP (**C**) prostate cells, as examined by flow cytometry after 24 and 48 h of treatment. The induction of growth arrest was affected by N-acetylcysteine (NAC) pretreatment. Cells were pretreated with 1.5 mM NAC for 1 h and then exposed to 0.4 mM arecoline for 24 and 48 h. Data are expressed as the mean ± SEM of triplicate experiments. Groups with different letters were significant (*p* < 0.05) in a given phase. In some groups, the standard error bars are too small to be seen. The symbols of “+” and “‒” were presented with or without addition of either arecoline or NAC, respectively.

**Figure 3 ijms-21-09219-f003:**
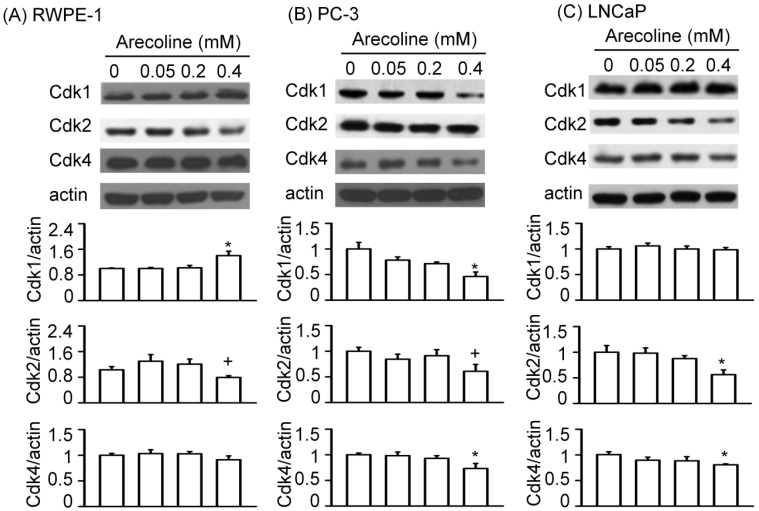
Differential effects of arecoline on proteins in the cyclin-dependent kinase (CDK) family between normal RWPE-1 (**A**) and cancerous PC-3 (**B**) and LNCaP (**C**) prostate cells after 24 h of treatment. Data are expressed as the mean ± SEM from triplicate experiments after Western blot analysis; each result was pooled from the data of four 10 cm culture plates. * *p* < 0.05 vs. the control. + *p* < 0.10 vs. the control.

**Figure 4 ijms-21-09219-f004:**
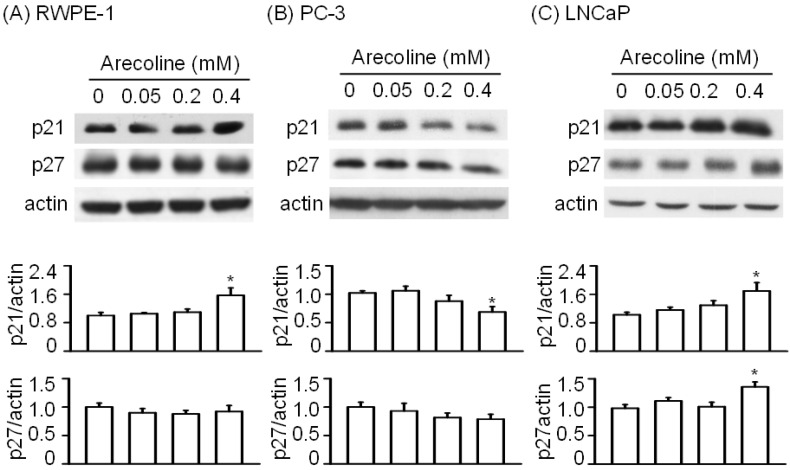
Differential effects of arecoline on proteins in the cyclin-dependent kinase inhibitor (CKI) family between normal RWPE-1 (**A**) and cancerous PC-3 (**B**) and LNCaP (**C**) prostate cells after 24 h of treatment. Data are expressed as the mean ± SEM from triplicate experiments after Western blot analysis; each result was pooled from the data of four 10 cm culture plates. * *p* < 0.05 vs. the control.

**Figure 5 ijms-21-09219-f005:**
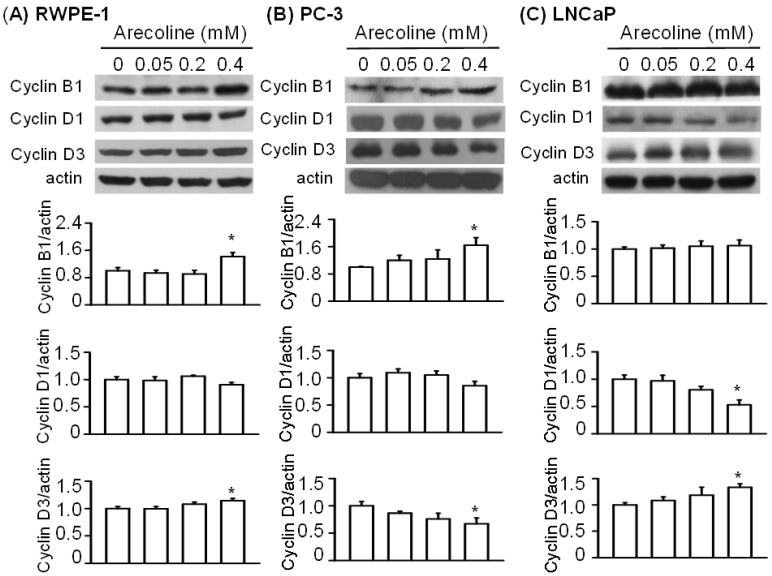
Differential effects of arecoline on the proteins in the cyclin family between normal RWPE-1 (**A**) and cancerous PC-3 (**B**) and LNCaP (**C**) prostate cells after 24 h of treatment. Data are expressed as the mean ± SEM from triplicate experiments after Western blot analysis; each result was pooled from the data of four 10 cm culture plates. * *p* < 0.05 vs. the control.

**Figure 6 ijms-21-09219-f006:**
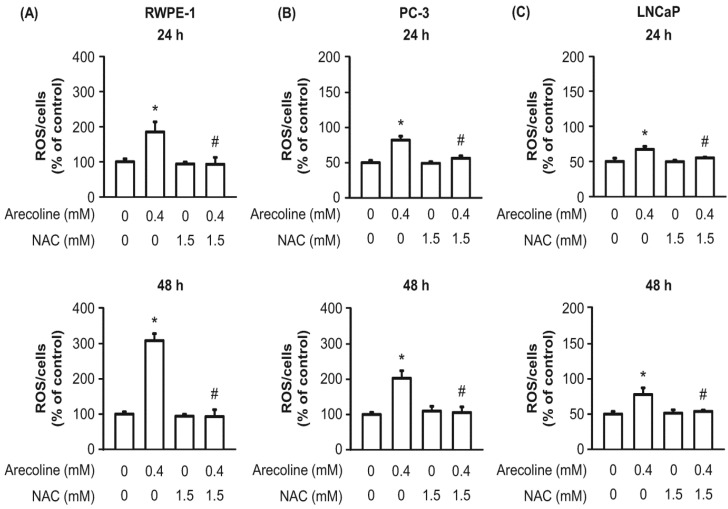
The effect of arecoline on the production of reactive oxygen species (ROS) was observed in normal RWPE-1 (**A**) and cancerous PC-3 (**B**) and LNCaP (**C**) prostate cells after 24 and 48 h of treatment. The induction of ROS production was reduced by pretreatment with N-acetylcysteine (NAC). Cells were pretreated with 1.5 mM NAC and then exposed to arecoline for 24 or 48 h. The ROS production was measured using the 2′,7″-dichlorofluorescein diacetate method. * *p* < 0.05 vs. the control; # *p* < 0.05 arecoline vs. arecoline + NAC.

**Figure 7 ijms-21-09219-f007:**
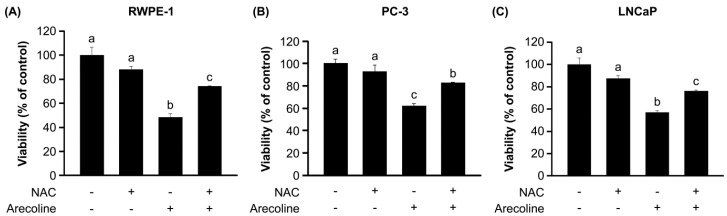
N-acetylcysteine (NAC) blocked the arecoline-induced changes in the viability of normal RWPE-1 (**A**) and cancerous PC-3 (**B**) and LNCaP (C) prostate cells. Cells were pretreated with NAC for 1 h and then exposed to 0.4 mM arecoline. After 48 h of treatment, we measured cell viability using an MTT assay. Data are expressed as the mean ± SEM from triplicate experiments. ^a–c^ Groups with different letters differ significantly (*p* < 0.05). The symbols of “+” and “‒” were presented with or without addition of either arecoline or NAC, respectively.

**Figure 8 ijms-21-09219-f008:**
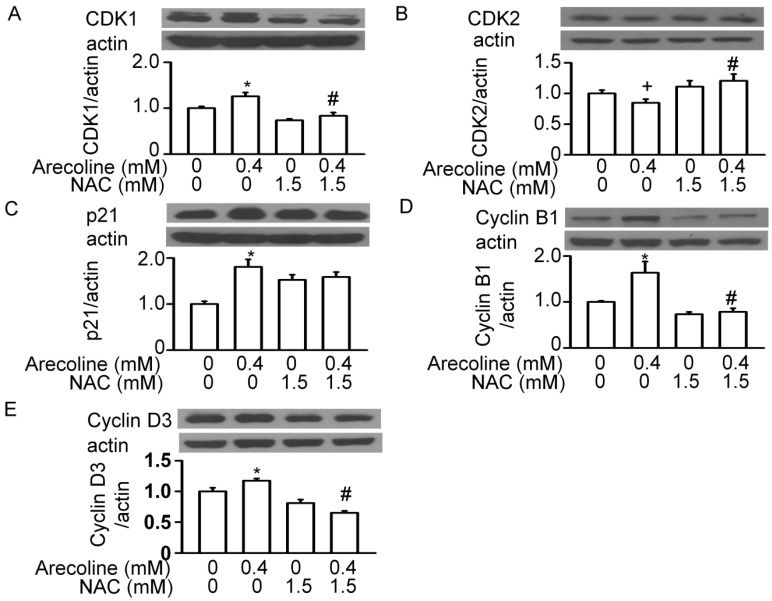
N-acetylcysteine (NAC) blocked the arecoline-induced alterations in the levels of cell cycle regulatory proteins, such as CDK1 (**A**), CDK2 (**B**), p21 (**C**), Cyclin B1 (**D**) and Cyclin D3 (**E**), in normal RWPE-1 prostate cells. Cells were pretreated with NAC for 1 h and then exposed to 0.4 mM arecoline. After 24 h of treatment, proteins were examined by Western blot analysis, and their levels were then expressed after normalization to actin expression. Data are expressed as the mean ± SEM from triplicate experiments; each result was pooled from the data of four 10 cm culture plates. * *p* < 0.05 vs. the control; + *p* < 0.05 vs. the control; # *p* < 0.05 arecoline vs. NAC + arecoline.

**Figure 9 ijms-21-09219-f009:**
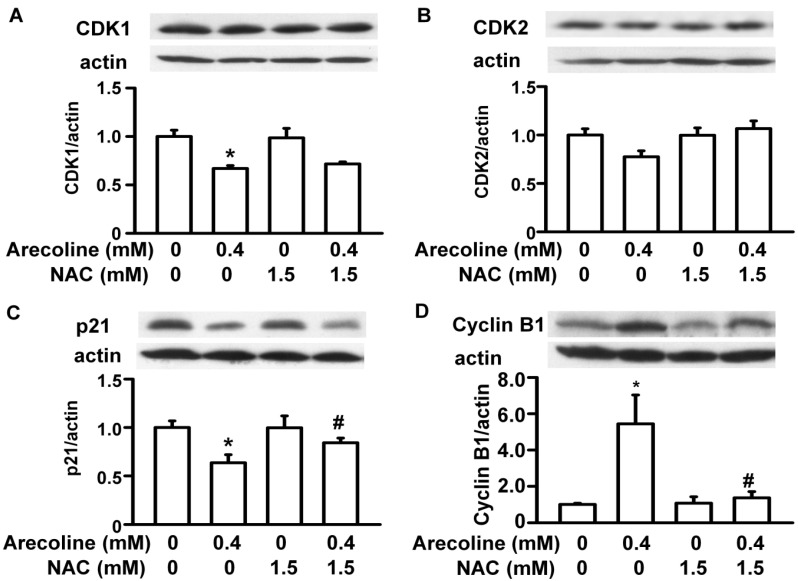
N-acetylcysteine (NAC) blocked the arecoline-induced alterations in the levels of cell cycle regulatory proteins, such as CDK1(**A**), CDK2(**B**), p21(**C**) and Cyclin B1(**D**), in androgen-independent human PC-3 prostate cancer cells. Cells were pretreated with NAC for 1 h and then exposed to 0.4 mM arecoline. After 24 h of treatment, proteins were examined by Western blot analysis and then expressed after normalization to actin expression. Data are expressed as the mean ± SEM from triplicate experiments; each result was pooled from the data of four 10 cm culture plates. * *p* < 0.05 vs. the control; # *p* < 0.05 arecoline vs. NAC + arecoline.

**Figure 10 ijms-21-09219-f010:**
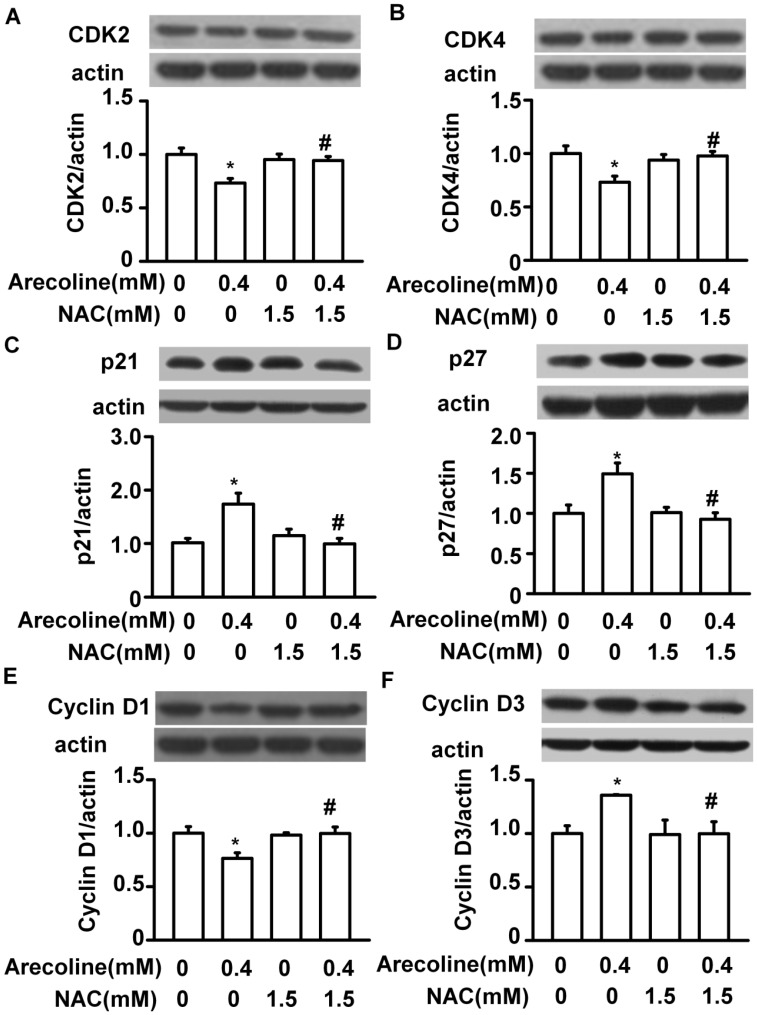
N-acetylcysteine (NAC) blocked the arecoline-induced alterations in the levels of cell cycle regulatory proteins, such as CDK2 (**A**), CDK4 (**B**), p21 (**C**), p27 (**D**), Cyclin D1 (**E**) and Cyclin D3 (**F**) in androgen-dependent human LNCaP prostate cancer cells. Cells were pretreated with NAC for 1 h and then exposed to 0.4 mM arecoline. After 24 h of treatment, the protein levels were examined by Western blot analysis and were then expressed after normalization to actin expression. Data are expressed as the mean ± SEM from triplicate experiments; each result was pooled from the data of four 10 cm culture plates. * *p* < 0.05 vs. the control; # *p* < 0.05 arecoline vs. NAC + arecoline.

**Figure 11 ijms-21-09219-f011:**
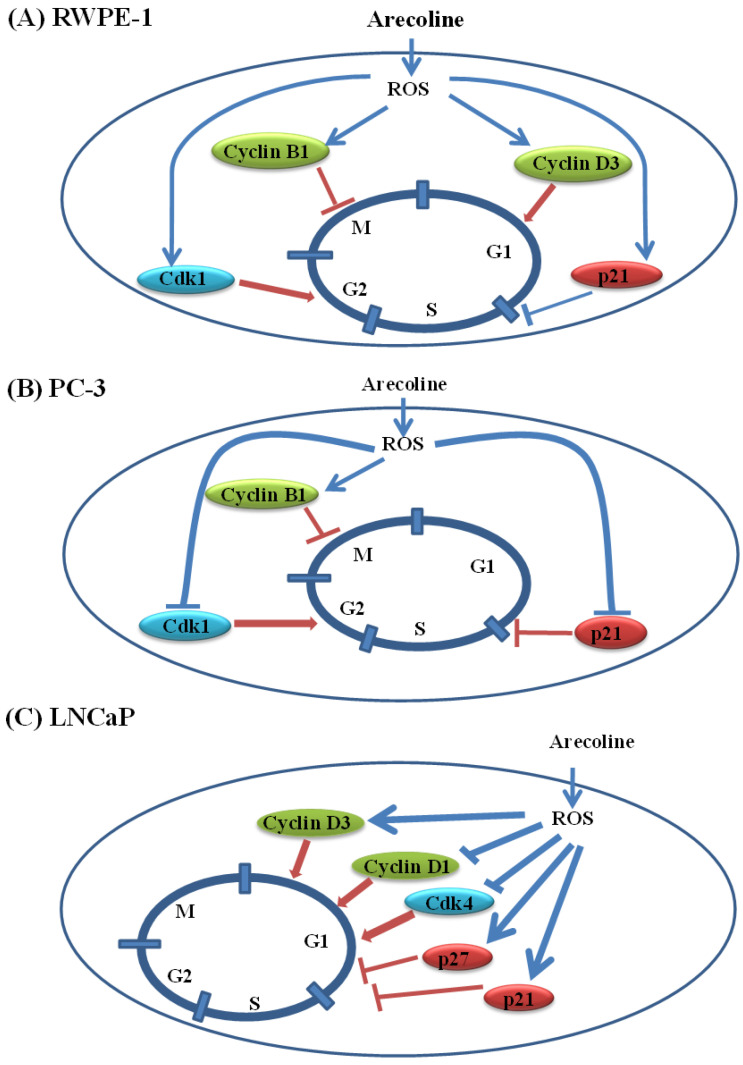
A proposed mechanism of the differential actions of arecoline on cell growth among RWPE-1 (**A**), PC-3 (**B**), and LNCaP (**C**) cells. The arecoline signaling was dependent on the cell cycle-controlling protein pathway, such as CDK1, CDK4, p21, p27, cyclin B1, or cyclin D3 proteins, in the reactive oxygen species (ROS)-dependent way. Thus, arecoline arrested RWPE-1 and PC-3 cells at the G2/M phase and LNCaP cells at the G0/G1 phase, leading to growth inhibition of all cells. The symbols of “→” and “┤” were presented with either stimulatory or inhibitory effect of arecoline on protein expression.

**Table 1 ijms-21-09219-t001:** Antibodies used in the experiments.

Name of Antibody	Species Raised in; Monoclonal or Polyclonal	Manufacturer, Catalog Number (#), and/or Name of Individual Providing the Antibody
Anti-β-actin	Rabbit; Monoclonal	Cell Signaling Technology, Inc. (Billerica, MA, USA), #12620
Anti-CDC2	Rabbit; Polyclonal IgG	Santa Cruz (Santa Cruz, CA, USA), sc-53
Anti-CDK2	Rabbit; Polyclonal IgG	Santa Cruz (Santa Cruz, CA, USA), sc-163
Anti-CDK4	Rabbit; Monoclonal	Cell Signaling Technology, Inc. (Billerica, MA, USA), #12790T
Anti-cyclin B1	Rabbit; Monoclonal	Cell Signaling Technology, Inc. (Billerica, MA, USA), #4138S
Anti-cyclin D1	Rabbit; Monoclonal	Cell Signaling Technology, Inc. (Billerica, MA, USA), #2978T
Anti-cyclin D3	Mouse; Monoclonal	Cell Signaling Technology, Inc. (Billerica, MA, USA), #2936T
Anti-p21	Rabbit; Monoclonal	Cell Signaling Technology, Inc. (Billerica, MA, USA), #2947T
Anti-p27	Rabbit; Monoclonal	Cell Signaling Technology, Inc. (Billerica, MA, USA), #3686T
Donkey anti-rabbit IgG-HRP	Donkey; Polyclonal	Santa Cruz (Santa Cruz, CA, USA), sc-2313
Goat anti-mouse IgG-HRP	Goat; Polyclonal	Santa Cruz (Santa Cruz, CA, USA), sc-2005

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
