# Peer review of "Betel Nut Arecoline Induces Different Phases of Growth Arrest between Normal and Cancerous Prostate Cells through the Reactive Oxygen Species Pathway"

_ijms, 2020, doi:10.3390/ijms21239219_

Round 1
Reviewer 1 Report
In this manuscript, Dr. Shih and colleagues focus their analysis on the properties of arecoline. They observe that arecoline has different effects on cell-cycle and on the expression of proteins involved in cell-cycle. Furthermore, the antioxidant N-acetylcysteine (NAC) blocks the arecoline-induced increase in reactive oxygen species production, decreases cell viability, and influences the cell-cycle.
This manuscript is interesting, but a lot of questions remain open:
- What is the target of arecoline? How does it exert these effects?
- The authors use three very different cellular lines, but sometimes the results are superimposible. How is it possible? On the contrary, why in LNCaP the results are different? In what these cell are different from the point of view of arecoline?
- The authors should introduce better guacine and arecaidine in the introdcution.
- the authors should add the plots of FACS analysis.
- Please, explain better line 124.
- Fig. 4A: from the western blot is not evident p21 increase.
- About fig. 5A: cyclin d3 is not evident the increase.
- The manuscript should be written in a more comprehensive and easier way.
Author Response
Responses to reviewer 1:
Thank you for your kind comments. We revised the text and added some details in the Materials and Methods as you suggested. Also, we added some descriptions in the Results and Discussion as you suggested. We must notify you that because of Reviewer 2’s suggestion to add a cartoon Fig. 11 for the summary of this article, we also made some changes in the number of figures and pages. We also added three paragraphs to the Discussion as you and Reviewer 2 suggested. My responses to your specific comments are as follows.
- What is the target of arecoline? How does it exert these effects?
Reply: We added one paragraph (Lines 280-291) in the Results and Discussion to discuss the target of arecoline.
- The authors use three very different cellular lines, but sometimes the results are superimposible. How is it possible? On the contrary, why in LNCaP the results are different? In what these cell are different from the point of view of arecoline?
Reply: As you and Reviewer 2 suggested, we added one paragraph (Lines 292-311) in the Results and Discussion to discuss the different effect of arecoline among three different cell lines.
- The authors should introduce better guacine and arecaidine in the introdcution.
Reply: We added the descriptions about guvacine and arecaidine (Lines 61-66) to the Introduction as you suggested.
- the authors should add the plots of FACS analysis.
Reply: We added the plots to the Supplemental Figure1A~C
- Please, explain better line 124.
Reply: We modified the descriptions (Lines 130-137) for clarity as you suggested.
- 4A: from the western blot is not evident p21 increase.
Reply: We replaced the Western blot with a new gel as you suggested.
- About fig. 5A: cyclin d3 is not evident the increase.
Reply: We replaced the Western blot with a new gel as you suggested.
- The manuscript should be written in a more comprehensive and easier way.
Reply: We modified the manuscript in a more comprehensive and easier way as you suggested.
Responses to reviewer 2:
We appreciate your kind comments. We revised the text, added a cartoon figure 11, and more descriptions in the Discussion as you suggested. We also create one paragraph to discuss the target of arecoline because of the Reviewer’s suggestion. We must notify you of this change. Thus, the number of references and the place of lines are changed in this reversion. My responses to your specific comments are as follows.
In response to the specific comments:
- This inhibition is somewhat surprising given that Saha et al. (2007, 2011) have reported that these same alkaloids act to increase cell proliferation in the prostate of rats. This needs to be discussed.
Reply: We created one paragraph next to the final paragraph (Lines 318-336) of the Results and Discussion to discuss the discrepancy of our study from Saha et al. report.
- Also, even though the author's studies appear to be well-conducted, they do not provide much in the way of interpretation of their results in terms of a coherent model or paradigm that would serve to unify their observations. Perhaps a diagram as the last figure would serve this purpose.
Reply: We added Figure 11 to the last figure as you suggested.
- The authors need to also present some rationale for the different effects of the alkaloids on the three cell types in terms of their androgen sensitivity and their difference from normal prostate epithelial cells.
Reply: As you suggested, we created one paragraph (Lines 286-317) to discuss the different effects of arecoline on the three cell types in terms of their androgen sensitivity and their difference from normal prostate epithelial cells.
Reviewer 2 Report
This manuscript reports the effects of three areca nut alkaloids on the growth of three prostate cell lines. The influence of these alkaloids on cell proliferations is attributed to their alterations in the expression of cyclins and CDKs in an ROS dependent manner. This inhibition is somewhat surprising given that Saha et al. (2007, 2011) have reported that these same alkaloids act to increase cell proliferation in the prostate of rats. This needs to be discussed. Also, even though the author's studies appear to be well-conducted, they do not provide much in the way of interpretation of their results in terms of a coherent model or paradigm that would serve to unify their observations. Perhaps a diagram as the last figure would serve this purpose. The authors need to also present some rationale for the different effects of the alkaloids on the three cell types in terms of their androgen sensitivity and their difference from normal prostate epithelial cells.
Author Response
Responses to reviewer 2:
We appreciate your kind comments. We revised the text, added a cartoon figure 11, and more descriptions in the Discussion as you suggested. We also create one paragraph to discuss the target of arecoline because of the Reviewer’s suggestion. We must notify you of this change. Thus, the number of references and the place of lines are changed in this reversion. My responses to your specific comments are as follows.
In response to the specific comments:
- This inhibition is somewhat surprising given that Saha et al. (2007, 2011) have reported that these same alkaloids act to increase cell proliferation in the prostate of rats. This needs to be discussed.
Reply: We created one paragraph next to the final paragraph (Lines 318-336) of the Results and Discussion to discuss the discrepancy of our study from Saha et al. report.
- Also, even though the author's studies appear to be well-conducted, they do not provide much in the way of interpretation of their results in terms of a coherent model or paradigm that would serve to unify their observations. Perhaps a diagram as the last figure would serve this purpose.
Reply: We added Figure 11 to the last figure as you suggested.
- The authors need to also present some rationale for the different effects of the alkaloids on the three cell types in terms of their androgen sensitivity and their difference from normal prostate epithelial cells.
Reply: As you suggested, we created one paragraph (Lines 286-317) to discuss the different effects of arecoline on the three cell types in terms of their androgen sensitivity and their difference from normal prostate epithelial cells.
Round 2
Reviewer 1 Report
For me, the authors improved the manuscript in particular with the discussion.
Now it is suitable for publication.